# Comparative Efficiency of Lutein and Astaxanthin in the Protection of Human Corneal Epithelial Cells In Vitro from Blue-Violet Light Photo-Oxidative Damage

Martina Cristaldi [1,†], Carmelina Daniela Anfuso [2,†], Giorgia Spampinato [1], Dario Rusciano [1,*] and Gabriella Lupo [2]

[1] Research Center, Sooft Italia SpA c/o Biologic Tower, University of Catania, 95123 Catania, Italy; martina.cristaldi@sooft.it (M.C.); giorgia.spampinato@sooft.it (G.S.)

[2] Department of Biomedical and Biotechnological Sciences, Biologic Tower, University of Catania, 95123 Catania, Italy; daniela.anfuso@unict.it (C.D.A.); gabriella.lupo@unict.it (G.L.)

[*] Correspondence: dario.rusciano@sooft.it; Tel.: +39-346-366-0348

[†] These authors contributed equally to this work.

**Abstract:** The aim of this study was to compare in vitro the protective and antioxidant properties of lutein and astaxanthin on human primary corneal epithelial cells (HCE-F). To this purpose, HCE-F cells were irradiated with a blue-violet light lamp (415–420 nm) at different energies (20 to 80 J/cm$^2$). Lutein and astaxanthin (50 to 250 µM) were added to HCE-F right before blue-violet light irradiation at 50 J/cm$^2$. Viability was evaluated by the CKK-8 assay while the production of reactive oxygen species (ROS) by the H2DCF-DA assay. Results have shown that the viability of HCE-F cells decreased at light energies from 20 J/cm$^2$ to 80 J/cm$^2$, while ROS production increased at 50 and 80 J/cm$^2$. The presence of lutein or astaxanthin protected the cells from phototoxicity, with lutein slightly more efficient than astaxanthin also on the blunting of ROS, prevention of apoptotic cell death and modulation of the Nrf-2 pathway. The association of lutein and astaxanthin did not give a significant advantage over the use of lutein alone. Taken together, these results suggest that the association of lutein and astaxanthin might be useful to protect cells of the ocular surface from short (lutein) and longer (astaxanthin) wavelengths, as these are the most damaging radiations hitting the eye from many different LED screens and solar light.

**Keywords:** blue-violet light; lutein; astaxanthin; ROS; corneal cells

## 1. Introduction

Wavelengths between 380 and 700 nm make the portion of the electromagnetic spectrum that is perceived by the human eye. The energy of electromagnetic radiations, including visible light, is a function of its wavelength, so shorter wavelengths are most energetic. The highest-energy portion of the visible spectrum is comprised between 380 and 500 nm, corresponding to blue-violet light [1]. The major source of blue-violet light (25–30%) is represented by sunlight. However, besides this natural source, emission peaks in the blue-violet light range may also come from artificial devices such as LED screens [2]. Many of us, in fact, are exposed every day to different sources of artificial lighting, such as white LED lamps, or LEDs used in flat screens of TVs, computer monitors, tablets and smartphones. These LED lamps are able to produce white light in three ways: (1) through the coupling of a diode with a phosphor, each emitting high- or low-frequency waves; (2) through the coupling of a diode that emits in the UV field with one or more phosphors; and (3) using several diodes that emit at different frequencies in the visible range [3]. These artificial light sources decay over time, mainly through a bleaching of phosphor, which is no longer able to absorb blue light, so that the emission of blue light increases over time [4]. We know today that ocular tissues are susceptible to damage from blue-violet light, both in the

back and in the front of the eye. Exposure of mice to common LEDs or blue light caused a significant increase in ROS production, the activation of inflammatory cytokines, and ultimately the death of photoreceptors in the retina. [5,6]. Although blue-violet light plays an essential role in color vision, excessive exposure to these wavelengths can have harmful effects due to the excessive production of ROS, which tend to damage photoreceptor cells and cells of the retinal pigmented epithelium (RPE) [7–10]. So far, the role of blue-violet light on the ocular surface has been the subject of few investigations. What is known is that the generation of ROS stimulated by blue-violet light can cause inflammation of the cornea and the apoptosis of corneal cells, ultimately leading to the development or aggravation of ocular surface dysfunctions, such as dry eye [11,12]. Lutein and astaxanthin are carotenoids belonging to the xanthophylls class, containing oxygen and being less hydrophobic with respect to carotenes [13]. Lutein, together with its stereoisomer zeaxanthin, is the only carotenoid present in the human retina [14], with the highest amount concentrated in the macula, where lutein dominates in the periphery, while zeaxanthin and meso-zeaxanthin become predominant in the center [15,16]. Astaxanthin is a red-colored metabolite of zeaxanthin and/or canthaxanthin, containing both hydroxyl and ketone functional groups. It is very common in marine organisms feeding on phytoplankton and microalgae, such as in salmonids, shrimps and lobsters, which thus acquire a pinkish-red color [17]. It is present in the retina of fishes, amphibians, reptiles and birds in oil droplets positioned in the photoreceptor inner segment, in front and apart from the outer segment where visual pigments are concentrated [18]. In comparison to other carotenoids, such as α- and β-carotene, lycopene and lutein, astaxanthin has shown a higher antioxidant activity in vitro [19]. Antioxidant carotenoids must be taken through the diet because they cannot be synthesized by higher organisms [20]. However, very often the amount of dietary carotenoids appears to be not sufficient for a good protection of eye structures, and several studies have pointed at the advantage of dietetic supplements to increase the concentration of carotenoids in the eye [21–23]. Dietary carotenoids mainly accumulate in the macular pigment of the retina, but they can be found also in other organs, including the skin [24]. Among human eye tissues, HPLC analysis has shown a relevant presence of dietary lutein also in the ciliary body, iris and lens, and detectable traces of lutein were also present in the cornea [25]. Recently, a topical formulation of lutein has been obtained by including lutein in a mixture of lipid nanoparticles and cyclodextrins. This allowed on the one hand an increased stability of lutein, and on the other hand an improved penetration of lutein into the cornea, and an increased residence time [26]. The protection against photo-oxidative stress exerted by carotenoids is due in part to their antioxidant activity and in part to their shielding effect on light radiations. Lutein and zeaxanthin, being stereoisomers, have almost superimposable profiles, with main absorption peaks around 460 nm. Astaxanthin has an absorption profile shifted towards the lower frequencies, with a main absorption peak at 492 nm [27]. In this study, we have developed a model of blue-violet light hazard with the newly described human corneal epithelial cell line HCE-F [28]. Epithelial cells plated in monolayers were irradiated with high energy light in the presence or the absence of lutein and astaxanthin either alone or in combination, and soon after intracellular production of free radicals and cell viability were measured, in order to test the xanthophylls' ability to shield the corneal surface from blue-violet light hazard.

## 2. Materials and Methods

### 2.1. Molecules

Lutein was purchased from Molekula Group (cat. no. 29291878) and astaxanthin from Sigma-Aldrich (St. Louis, MO, USA, cat. no. 38028884).

### 2.2. Cell Culture

HCE-F cells were isolated from the human cornea of a donor patient after keratoplastic surgery [28]. Cells were cultivated in DMEM-F12 Advanced (ATCC, cat. no. 12634010) and supplemented with 1% penicillin/streptomycin, 2% FBS (Sigma-Aldrich, St. Louis,

MO, USA, cat. no. F7524) and specific corneal epithelial growth factors (ATCC, cat. no. PCS-700-040) at 37 °C in a humidified atmosphere containing 5% $CO_2$.

### 2.3. LED Light Source

A 25 Watt LED light lamp, emitting at 415–420 nm was purchased from Taoyuan Electron (Hk) Ltd (Hong Kong, China). The irradiance of the lamp was measured using a power-meter (Thorlabs, Bergkirchen, Germany) when the lamp was placed 5 cm over the cell layer.

### 2.4. Blue-Violet Light Irradiation

HCE-F cells were seeded at a density of $2 \times 10^4$ cells per well into 96-well plates and left to adhere overnight at 37 °C in a humidified atmosphere containing 5% $CO_2$. After 24 h, the culture medium was replaced with serum-free medium (SFM) with or without lutein and astaxanthin, and cells were placed under the blue-violet light LED lamp at a distance of 5 cm for different times so to attain 20, 50, 80 J/cm$^2$ [29]. Controls were left in the dark. After each exposure, ROS intracellular content and cell viability were measured as described below.

### 2.5. Measurement of Intracellular ROS

HCE-F cells were seeded in culture medium at a density of $2 \times 10^4$ cells per well into 96-well plates. The day after, culture medium was replaced by SFM with 0, 50, 100, 250 μM lutein, 0, 50, 100, 250 μM astaxanthin or a mix of 100 μM lutein and 100 μM astaxanthin, and then cells were exposed for 30 min to blue-violet light at a final dose of 50 J/cm$^2$. The levels of intracellular ROS were analyzed using 2′,7′-dichlorodihydrofluorescein diacetate (H2DCFDA, Life Technologies, Invitrogen™, Waltham, MA, USA 02451; cat. no. D-399) [30,31] according to the manufacturer's instructions. Briefly, adherent HCE-F cells were washed once with SFM and incubated with 10 μM H2DCFDA (loading buffer) at 37 °C for 30 min. The loading buffer was then removed and the cells returned to SFM. The fluorescence intensity (λex = 492 nm, λem = 517 nm) was measured with the Varioskan™ (Thermo Fisher Scientific, Waltham, MA, USA 02451).

### 2.6. Cell Viability

After ROS measurement, cell viability with the soluble CCK-8 assay (Sigma-Aldrich, St. Louis, MO, USA, cat. no. 96992) [32] was performed in the same plate. Briefly, 10 μL of CCK-8 solution in 100 μL of serum-free medium were added to each well for 1.5 h at 37 °C in a humidified atmosphere containing 5% $CO_2$. At the end of the incubation time, the absorbance was measured at 450 nm by the plate reader Synergy 2 (BioTek, Thermo Fisher Scientific, Waltham, MA, USA). O.D. values were used to evaluate cell survival and to normalize ROS levels.

### 2.7. Analysis of Apoptotic Cells

First, $2 \times 10^4$ HCE-F cells per well were seeded into a 96-well plate. The day after, culture medium was replaced by SFM, with or without the presence of lutein and astaxanthin either alone, or in association, each one at a final concentration of 100 μM. Adherent cells were exposed 30 min to blue-violet light at a final dose of 50 J/cm$^2$. After treatment, 2 drops/mL of CellEvent™ Caspase-3/7 Green Detection Reagent (Thermo Fisher Scientific, cat. no. R37111, Waltham, MA, USA) to mark apoptotic nuclei and 2 mg/mL Hoechst 33342 Thermo Fisher Scientific, cat. no. H35 70, Waltham, MA, USA) to label total nuclei were added to each well, and the plates incubated for 30 min at 37 °C in a humidified atmosphere in the presence of 5% $CO_2$. Fluorescent cells were observed using a fluorescence microscope LEICA DMi4000, and representative pictures were acquired with the integrated camera and analysed measuring intensity of green and blue fluorescence through imaging software Image J [33]; the analysis was reported as the green/blue fluorescence intensity ratio.

*2.8. Western Blot Analyses of Cell Lysates*

After treatments with blue-violet light, as described above, in the presence of lutein, astaxanthin either alone, or in association at a final concentration of 100 μM, cell monolayers were washed twice with PBS and harvested mechanically using a cell scraper. Cells were centrifuged at 1000 rpm for 5 min at 25 °C, and pellets were lysed in RIPA buffer (Calbiochem-Merck, cat. no. 20188) supplemented with protease and phosphatase inhibitor cocktails (Protease Inhibitor Cocktail Set III EDTA-Free, Calbiochem-Merck, Cat. No. 539134; Phosphatase Inhibitor Cocktail 2, Sigma-Aldrich, cat. no. P5726, St. Louis, MO, USA; Phosphatase Inhibitor Cocktail 3, Sigma-Aldrich, cat. no. P0044, St. Louis, MO, USA) by incubation for 30 min on ice. Extracts were clarified by centrifugation at 13,000 rpm for 20 min at 4 °C and proteins in the supernatant quantitated by the BCA protein assay (BCA kit assay, Santa Cruz biotechnology, cat. no. 10410, Santa Cruz, CA, USA). For Western blot analysis, 25 μg of proteins were loaded onto 4–20% polyacrylamide gel (Mini-PROTEAN® TGXTM Precast Protein Gels, cat. no. 4561096, Bio-Rad Laboratories, Segrate, Italy) followed by electrotransfer to nitrocellulose membranes (Trans-Blot Turbo Mini 0.2 μm nitrocellulose, cat. no. 1704158, Bio-Rad Laboratories, Italy). Membranes were then incubated overnight at 4 °C with primary antibodies (all diluted 1:1000): mouse monoclonal anti-Nrf2 antibody (Abcam, cat. no. ab89443, Cambridge, UK); rabbit polyclonal anti-Keap1 (Abcam, cat. no. ab 139729, Cambridge, UK); mouse monoclonal anti-Heme Oxygenase 1 antibody (Abcam, cat. no. ab13248, Cambridge, UK); rabbit monoclonal anti-Phospho-Akt (Ser473) (D9E) XP® (Cell Signaling Technology, cat. no. 4060, Danvers, MA, USA); rabbit polyclonal anti-Akt (Cell Signaling Technology, cat. no. 9272, Danvers, MA, USA) and rabbit monoclonal anti-GAPDH (Cell Signaling, Leiden, The Netherlands, Cat. No. 2118). Further incubation with IgG-HRP-conjugated anti-rabbit or anti-mouse secondary antibodies (Amersham, GE Healthcare, Bloomington, IL, USA, Cat. no. NA934V) diluted 1:2000 was completed for 1 h at room temperature. The immune complexes were detected by enhanced chemiluminescence (ECL Super-SignalTM West Dura Extended Duration Substrate, Thermo Fisher Scientific, cat. no. 34075, Waltham, MA, USA) with the Chemi-DocTM Touch Imaging System (Bio-Rad, Hercules, CA, USA). The intensity of protein bands was quantitated by ImageJ Software [33].

*2.9. Statistics*

All experiments were run at least in triplicates and repeated not less than three times. Statistical significance (set at $p < 0.05$) was evaluated by one-way ANOVA, followed by Tukey's test, using the Graph-pad version 8 software.

## 3. Results

*3.1. Blue-Violet LED Dose-Effect*

Cell viability and the ROS levels of HCE-F cells were evaluated after irradiation with the blue-violet LED lamp at 0, 20, 50 and 80 J/cm$^2$ (Figure 1). Survival of HCE-F progressively decreased from 82% at an irradiation potency 20 J/cm$^2$ to 23% when the irradiation potency was set at 80 J/cm$^2$ (Figure 1A). In parallel, the amount of ROS increased in a directly proportional way to the irradiation energy, starting from a 16% elevation at the lower irradiation energy and up to 20 times more at the higher irradiation potency (Figure 1B).

*3.2. Dose-Effect Response of Lutein and Astaxanthin*

The shielding and anti-oxidant power of different doses (50, 100 and 250 μM) of lutein and astaxanthin in this photo-stress model was tested at the irradiation energy of 50 J/cm$^2$, which gave an intermediate response in terms of toxicity and ROS production (Figure 1). These doses have been chosen solely in function of their safety and tolerability on HCEF cells and do not have any relationship with doses given in vivo, either as topical [15] or oral treatments [34,35], due to the very different pharmacodynamics of the two systems. In these experiments, in the absence of any protection, HCEF cell viability decreased

by 74% (Figure 2A), and ROS increased by 391% (Figure 2B). In the presence of lutein, a full recovery of viability could be observed at the highest doses of 100 and 250 μM (Figure 2A), and all three doses returned the ROS level similar to control values (Figure 2B). Astaxanthin was less efficient than lutein in protecting cell viability (Figure 2A), despite a significant reduction of ROS, which was however less relevant than what obtained with lutein (Figure 2B).

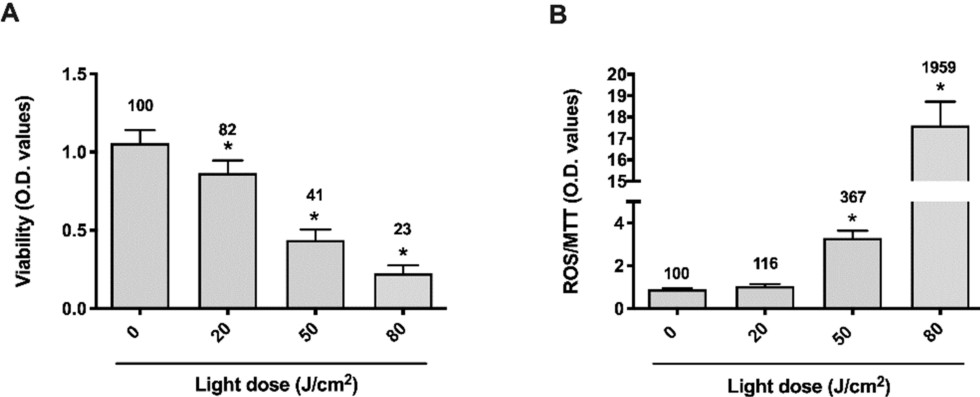

**Figure 1.** (**A**) Cell viability after 0, 20, 50 and 80 J/cm$^2$ blue-violet light irradiance was evaluated by the CKK-8 assay and reported as mean ± SD of O.D values. (**B**) H2DCFDA assay estimated intracellular ROS content after 0, 20, 50 and 80 J/cm$^2$ of blue-violet light irradiance; O.D. values were normalized to the respective viability O.D. values. Percent values are indicated over each bar. The experiment has been run with triplicates per each point. * $p < 0.05$ vs. 0 J/cm$^2$. One-way ANOVA followed by Tukey's test.

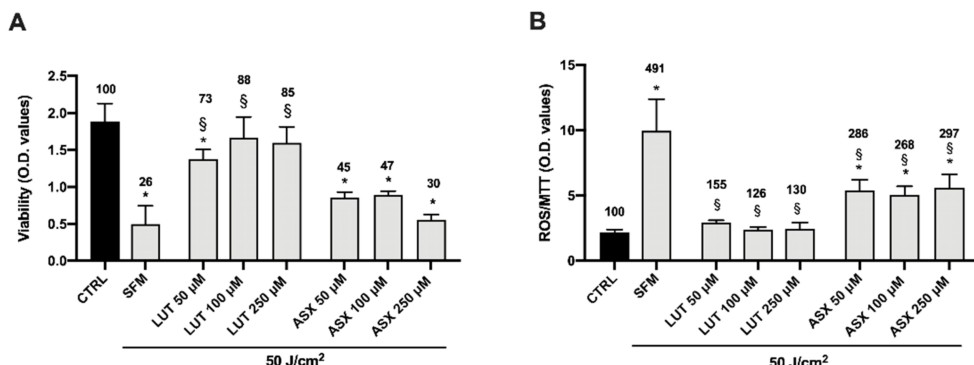

**Figure 2.** (**A**) Cell viability after 50 J/cm$^2$ blue-violet light irradiance in presence of lutein (LUT) 50–250 μM, and astaxanthin (ASX) 50–250 μM was evaluated by the CKK-8 assay and reported as mean ± SD of O.D. values. (**B**) H2DCFDA assay estimated intracellular ROS content in HCE-F cells irradiated at 50 J/cm$^2$; O.D. values were normalized to the respective viability O.D. values. Percent values are indicated over each bar. The experiment has been run with triplicates per each point. SFM: Serum Free Medium * $p < 0.05$ vs. CTRL; § $p < 0.05$ vs. SFM. One-way ANOVA followed by Tukey's test.

### 3.3. Cooperative Effect of Lutein and Astaxanthin

To identify a potential additional protective effect of lutein and astaxanthin, HCE-F cells were incubated with each molecule either alone at a concentration of 100 μM, or mixed together (100 μM of each), and immediately exposed at a blue-violet light dose of 50 J/cm$^2$ (Figure 3). As expected, the absence of any protection resulted in a 79% decrease in cell viability (Figure 3A) and a 394% increase in ROS production (Figure 3B). The single molecules alone confirmed the superiority of lutein in the protective effect, with only a 15% reduction in viability and no significant increase in ROS, while astaxanthin alone still resulted in a 47% loss of cell viability (Figure 3A) with a 117% increase of ROS production

(Figure 3B). The mix of the two resulted to be no better than 100 μM lutein alone in protecting cell viability (Figure 3A) or decreasing the ROS production (Figure 3B).

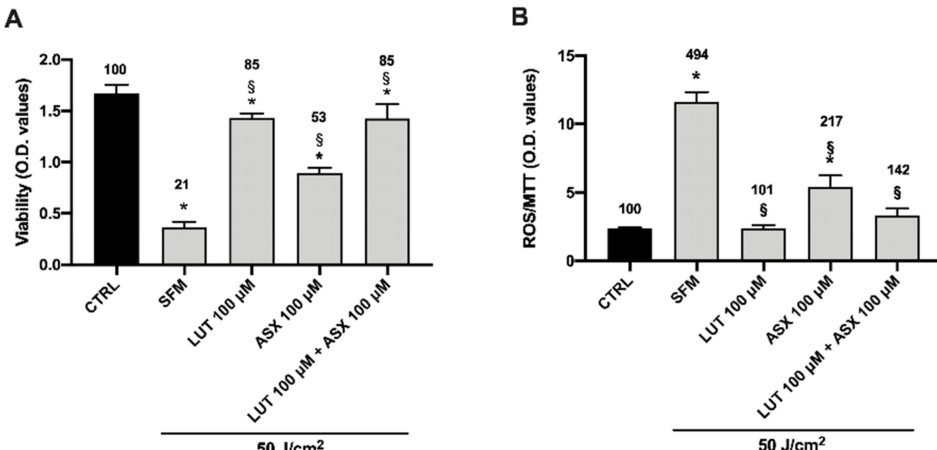

**Figure 3.** (**A**) Cell viability after 50 J/cm$^2$ blue-violet light irradiance in presence of 100 μM lutein (LUT) and 100 μM astaxanthin (ASX), alone and mixed, was evaluated by the CKK-8 assay and reported as mean ± SD of O.D values. (**B**) Intracellular ROS content performed by H2DCFDA assay; O.D. values were normalized to the respective viability O.D. values. Percent values are indicated over each bar. The experiment has been run with triplicates per each point. SFM: Serum Free Medium * $p < 0.05$ vs. CTRL; § $p < 0.05$ vs. SFM. One-way ANOVA followed by Tukey's test.

*3.4. Cell Death Induced by Blue-Violet Light Exposure in HCE-F Cells and Protective Role of Antioxidants*

Blue-violet light triggered a dramatic apoptotic cell death after the cells were exposed for 30 min to a final dose of 50 J/cm$^2$. Cells irradiated in the absence of antioxidant protection underwent massive apoptosis (roughly 17 times higher than control values, Figure 4). In the presence of 100 μM lutein, 100 μM astaxanthin or a mix of these two, we observed a significant decrease of apoptotic cells, which returned to values closer to control wells with a trend, although not significant, of a lesser protection by astaxanthin with respect to lutein.

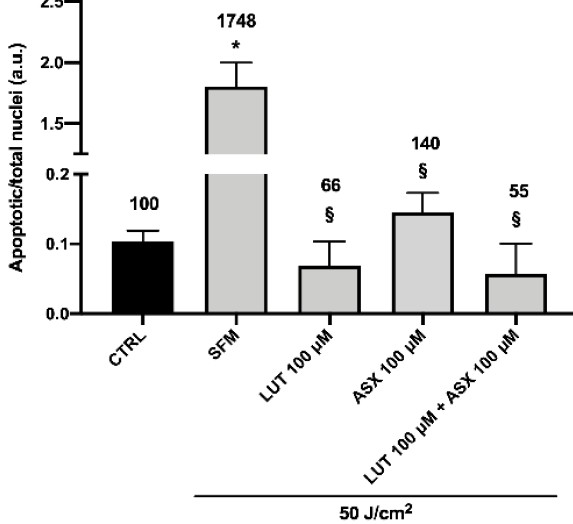

**Figure 4.** Ratio of apoptotic nuclei vs. Hoechst 33,342-stained nuclei evaluated in three different representative fields are shown. Relative percent numbers with respect to untreated control (CTRL) are shown on top of each bar. The experiment has been run with triplicates per each point. SFM: Serum Free Medium * $p \leq 0.05$ vs. CTRL; § $p < 0.05$ vs. SFM. One-way ANOVA followed by Tukey's test.

### 3.5. Western Blot Analysis of Endogenous Antioxidant Markers

The effect of photooxidative stress on the expression of endogenous antioxidant markers was evaluated soon after the treatment of HCEF cells with blue-violet light, in the absence or the presence of the xanthophylls lutein and/or astaxanthin. Figure 5A,C shows the behavior of the Nuclear factor erythroid 2-related factor 2 (Nrf-2), a transcription factor regulating the expression of genes involved in oxidative stress response and drug detoxification against toxic and oxidative insults [36]. Nrf-2 appears as a double band (likely representing the cytoplasmic—the lower band—and the nuclear—the upper band—form of Nrf-2) [37] dramatically fading under blue-violet light irradiation (55% of reduction). The presence of lutein or astaxanthin alone had a mild, non-significant effect on its expression, and only the combination of the two was able to operate a significant rescue of Nrf-2 expression, at 75% of control values. The activity of Nrf-2 is controlled by the protein Keap-1, which works as a suppressor of Nrf-2 when cells are not under stress [38]. Accordingly, photooxidative stress triggers a 43% decrease of Keap-1, which is further decreased in the presence of the two xanthophylls, down to 27% of control values when both were present (Figure 5B,D), thus liberating more active Nrf-2 to induce endogenous antioxidant defense. In fact, the ratio Nrf-2/Keap-1 progressively increased in the presence of lutein or astaxanthin to reach an almost 5-times increase in the presence of their association (Figure 5E). The expression of Heme-oxygenase-1 (HO-1) is directly regulated by Nrf-2. In fact, its expression pattern after photooxidative stress closely follows that observed for Nrf-2, with a 34% reduction in the absence of any protective molecular shield and with a full rescue when lutein and/or astaxanthin were present (Figure 6A,C). The PI3K/AKT pathway is essential for the regulation of the Nrf-2/HO-1 axis [39]. Indeed, Figure 6B,D show a dramatic (83%) downregulation of the activated form of AKT (pAKT), with only a modest, but significant, rescue by the presence of the xanthophylls, however with still a 60% reduction in the presence of the association.

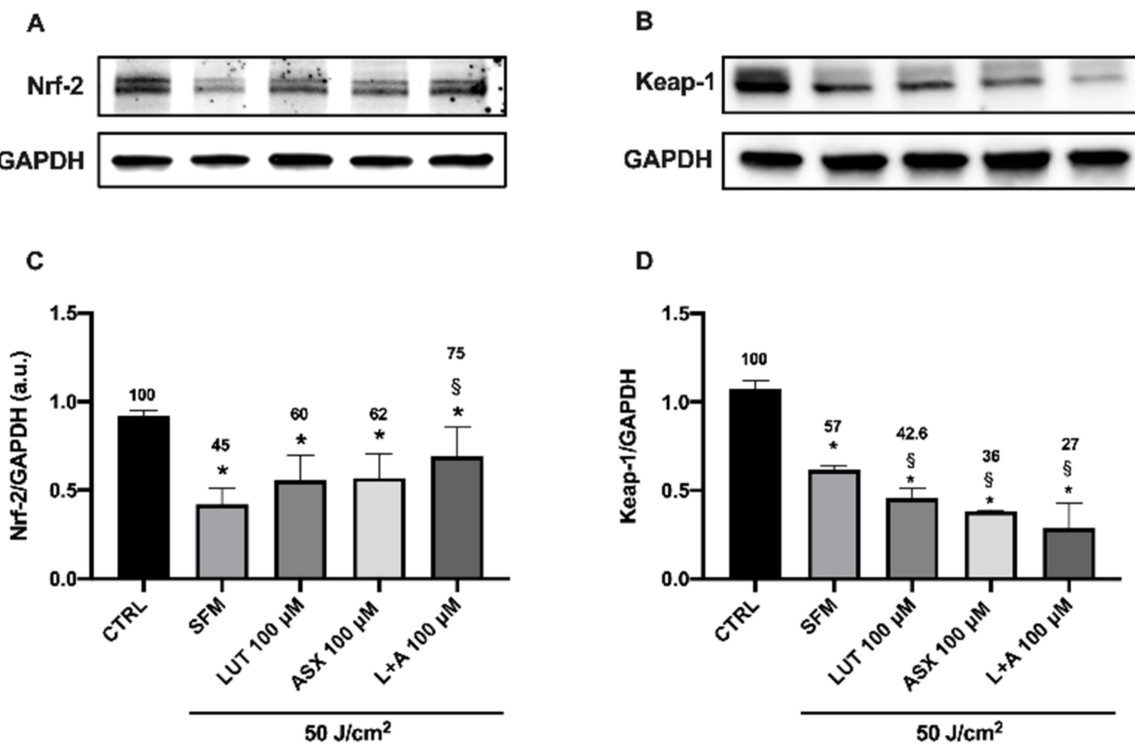

**Figure 5.** *Cont.*

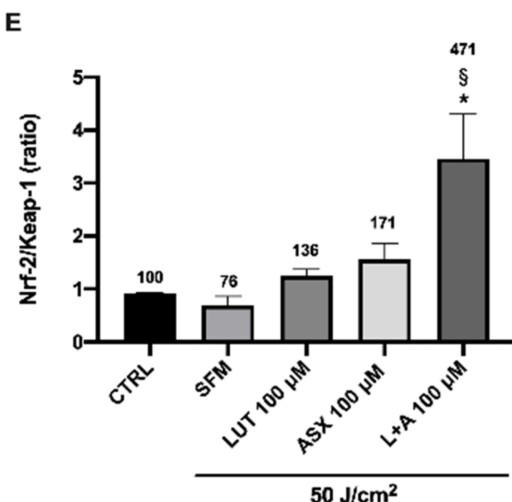

**Figure 5.** Western blot analysis of protein extracts after blue-violet irradiation (50 J/cm$^2$) in the absence or in presence of 100 μM LUT and 100 μM ASX either alone or in association. Protein expression of Nrf-2 and Keap-1 are reported in panels (**A**,**B**) and the respective quantitative analysis in panels (**C**–**E**), in which the corresponding percentage values are indicated at the top of each bar. Each bar represents the average value ± SD of three different experiments. SFM: Serum Free Medium. * $p < 0.05$ vs. CTRL; § $p < 0.05$ vs. SFM (irradiated cells, without any protection). One-way ANOVA, followed by Tukey's test.

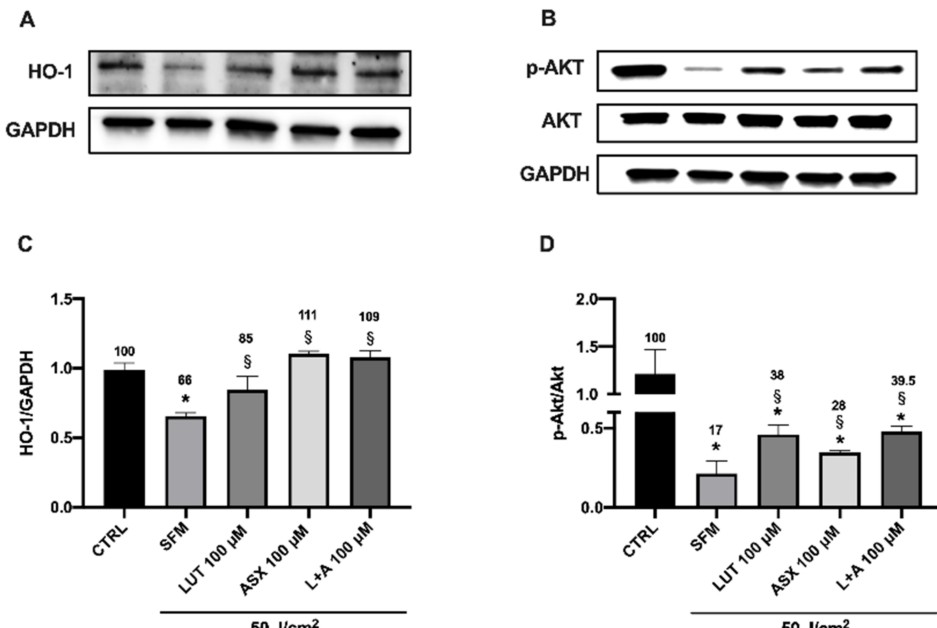

**Figure 6.** Western blot analysis of protein extracts after blue-violet irradiation (50 J/cm$^2$) in the absence or in the presence of 100 μM LUT, 100 μM ASX either alone or in association. Protein expression of HO-1 and AKT are reported in panels (**A**,**B**) and the respective quantitative analysis in panels (**C**,**D**), in which the corresponding percentage values are indicated at the top of each bar. Each bar represents the average value ± SD of three different experiments. SFM: Serum Free Medium. * $p < 0.05$ vs. CTRL; § $p < 0.05$ vs. SFM (irradiated cells, without any protection). One-way ANOVA, followed by Tukey's test.

## 4. Discussion

In this study, we have taken advantage of the two characterizing properties of the xanthophylls lutein and astaxanthin, i.e., to be strong antioxidants and ROS scavengers, and

to shield from low wavelength irradiation, even though with a different absorption pattern). In fact, our experimental protocol was more focused on the shielding effect because we evaluated apoptosis and viability immediately after the irradiation, and cells were not allowed a recovery time, during which the antioxidant efficacy of the xanthophylls could have been more evident. Therefore, the reduction of ROS observed in Figures 2B and 3B is likely mainly contributed by this shielding effect. In turn, this might explain why lutein appears more efficient than astaxanthin, since its radiation absorbance is higher in the blue-violet light interval used in our irradiation protocol (415–420 nm). Consequently, also cell viability after irradiation was higher with the lutein rather than with the astaxanthin shield (Figures 2A and 3A), despite the superiority of astaxanthin in terms of antioxidant potential [40]. A similar pattern was also observed for apoptotic nuclei (Figure 5). Lutein and its mix with astaxanthin appeared slightly more efficient than astaxanthin alone in blunting apoptosis induction by blue-violet light irradiation. A moderate light exposure is usually not dangerous to the eye, which contains enough endogenous anti-oxidants to cope with this amount of irradiation. However, when the radiation is more intense and/or prolonged over time, such as it may happen to people in the snow, or on the sea during a clear and sunny day, it can cause temporary or even permanent blindness [41]. Furthermore, the architecture of the outer blood-retinal barrier (formed by RPE cells) can also be altered by exposure to white LED light, as shown by the disorganization of the actin cytoskeleton, the induction of endoplasmic reticulum stress response and finally the activation of apoptosis in treated rats [42]. Similar findings were reported also after exposure of mice to white LED light. Light exposure increased ROS concentration in RPE cells and concomitantly disrupted the staining patterns of tight and adherents junctions, together with the actin cytoskeleton [43]. In humans, the amount of working time under bright sunlight seems to be a risk factor for the development of age-related macular degeneration. Therefore, preventive measures, able to shield from heavy sunlight exposure, should be started early in life to prevent the development of AMD later in life [44,45]. Besides, the eye surface may also suffer from photo-oxidative damage, and a battery of enzymatic and non-enzymatic antioxidants are present in the anterior segment of the eye to protect its structure from such damage [46]. Human corneal epithelial cells in vitro are damaged and lose viability after exposure to blue light, and such damage could be attenuated by blue light-filtering lenses [47]. Along the same line, Marek and colleagues demonstrated that photooxidative damage was worsened in presence of hyperosmolar stress (as it happens in dry eye patients), thus resulting in increased inflammation and disturbed mitochondrial membrane potential, finally turning on the cell antioxidant response based on the glutathione system. Therefore, it was suggested that patients with dry eye disease could be even more sensitive to the toxicity of blue-light [48]. In fact, from a clinical perspective, it has been reported that the use of spectacles equipped with lenses able to block 50% of blue light could improve visual acuity in dry eye patients [49]. Interestingly, the irradiation source is also critical for the toxic effects on the eye. The radiation produced from organic light-emitting diodes (OLEDs) has been shown to be less toxic to ocular surface and retinal cells than the light coming from common light-emitting diodes (LEDs), most likely because of a higher blue-wavelength energy irradiated by LEDs [50]. Finally, in a mouse model, it has been shown that over-exposure to blue-violet light could cause oxidative damage and the apoptosis of corneal cells, leading to increased inflammation of the ocular surface and eventually eye dryness [12]. Intriguingly, the toxic effects of irradiation by high frequency blue light could be counteracted by lower frequency red light. It has been shown that red light irradiation stimulates mitochondrial function and increases the proliferation of human corneal epithelial cells (HCE2) in vitro, thus suggesting a possible prophylactic treatment to protect the ocular surface of dry eye patients [51].

Photo-oxidative cellular damage is closely related to a dramatic increase of ROS. Therefore, the protective efficacy of the two well-known eye-specific natural antioxidants, lutein and astaxanthin, has been here investigated. Lutein is a carotenoid with a chemical structure that gives the molecule the ability to absorb blue-violet light somewhat more efficiently

than astaxanthin [52]. It has been demonstrated both in vitro and in vivo that lutein can protect the retina not only from photo-oxidative damage but also from inflammation by reducing cytokine expression levels and oxidative stress-induced apoptosis in photoreceptors [53–55]. A double-blind, placebo-controlled clinical study investigated the effects of lutein and zeaxanthin on glare disability, showing that daily supplementation with these two xanthophylls resulted in their significant increase in serum levels and in the macular region, with a parallel improvement in chromatic contrast and recovery from photostress [56]. Astaxanthin is another carotenoid, found in *Haematococcus pluvialis*, *Chlorella zofingiensis*, *Chlorococcum* and *Phaffia rhodozyma*, having anti-tumor, anti-diabetic and anti-inflammatory properties [17]. Astaxanthin, having a structure similar to lutein, can also absorb the short visible wavelengths, even though with a somewhat lower efficiency with respect to lutein. Nonetheless, astaxanthin may protect the retina from photodamage by attenuating the apoptosis of retinal ganglion cells and reducing oxidative stress [57,58]. In fact, it is known that astaxanthin may inhibit blue-violet light LED-induced cell apoptosis and exert its protective activity through the activation of the Nrf2-ARE pathway, thus increasing the expression of phase II antioxidant enzymes such as NADPH quinine oxidoreductase 1 (NQO1) and heme-oxygenase-1 (HO-1) [59–61]. The protective role against the toxicity of blue-violet light on human corneal epithelial cells of increased expression of HO-1, Prx-1, CAT and SOD-2 and the consequent attenuation of ROS production by some plant extracts has been shown in a recent paper by Lee and collaborators [62]. Moreover, it has been shown that both the oral and topical administration of astaxanthin may effectively protect corneal cells from apoptosis and significantly reduce oxidative stress following UV-induced photokeratitis damage [63,64]. In this study, we have focused our attention on the Nrf-2 pathway, which is known to be involved in the regulation of endogenous antioxidant defense also after blue light irradiation. In fact, the long-term exposure (24 to 48 h) of murine photoreceptor cells (661W) in vitro to low-intensity blue light was reported to increase ROS production and induce Nrf-2 expression [29,65]. In the same model system, astaxanthin was shown to induce the Nrf-2 protection system and decrease blue light cell damage [61]. Under our experimental conditions of acute photooxidative damage (50 J/cm$^2$ for 30 min), we observed a dramatic increase of ROS, a relevant decrease in cell viability and a parallel decrease of Nrf-2 expression (Figures 1 and 5). The presence of lutein and astaxanthin blunted ROS increase, thus improving cell viability and rescuing Nrf-2 expression (Figures 2, 3 and 5). The likely explanation of these data on Nrf-2—opposite to what shown in the previously cited studies—could rely on the observation that when an acute oxidative stress is given in the absence of an appropriate antioxidant defense (such as in our case), there is a sudden loss of cell viability due to mitochondrial toxicity as detected by the MTT assay [66], thus leading to a decrease of Nrf-2 expression. However, if a milder oxidative stress is given over a sufficiently wide time frame (as in the previous referenced studies on 661W cells), the Nrf-2 pathway is induced, and an increase of Nrf-2 expression can be observed. In either case, the shielding effect of lutein and astaxanthin, combined with their intrinsic antioxidant properties, may result in the recovery of the original situation: in our case, in a recovery of cell viability and Nrf-2 expression. Similar in vitro results were also reported for a human corneal epithelial cell line treated by hyperosmolar stress and for a rabbit corneal cell line treated by hyperglycemic stress, each treatment resulting in a decrease of cell viability and a concomitant reduction of Nrf-2 expression. In these cases, the decrease of both cell viability and Nrf-2 expression could be prevented by pretreatment with the antioxidant edaravone, in the case of hyperosmotic stress [67], or by dimethyl-fumarate, in the case of hyperglycemic stress [68]. An additional consideration is that Nrf-2 response to oxidative stress could be differently modulated in epithelial (a decrease is observed) and neuronal (an increase is reported) cells. A similar situation has been reported for scleral or choroidal fibroblasts, in which atropine, respectively, induced an increase or a decrease of collagen production [69]. Most important, in our opinion, is the behavior of Keap-1. This protein binds Nrf-2 and induces its degradation, thus preventing its functioning. Oxidants disrupt this interaction so that more Nrf-2 is free to activate the

DNA transcription of antioxidant genes [38,70]. In our study, we found a relevant decrease of Keap-1 expression (Figure 5B), unrelated to the presence of xanthophylls and cell survival. Such low levels of Keap-1, significantly lower in the presence of both xanthophylls, implies that more Nrf-2 can be available to induce endogenous antioxidant protection, thus contributing to enhanced cell viability. In particular, the ratio Nrf-2/Keap-1 is the highest when both astaxanthin and lutein were present together to shield HCEF cells from blue-violet light (Figure 5E). Nrf-2 also directly regulates HO-1 expression, as it is apparent from the expression pattern of HO-1 in Figure 6A [71]. The Nrf-2/HO-1 signaling pathway is involved in several functions, such as calcium regulation, mitochondrial oxidative stress, apoptosis and autophagy [72]. Not unexpectedly, pAKT—the activated form of AKT—also follows the expression pattern of Nrf-2 (Figure 6B). In fact, the activation by phosphory-lation of AKT resides upstream Nrf-2 and is required to induce its expression [73]. Our results show for the first time a direct comparison between lutein and astaxanthin in the protection of human corneal epithelial cells from blue-violet light damage and indicate a higher efficacy of lutein over astaxanthin, which might be correlated to the different absorption spectrum of the two molecules, with astaxanthin showing an absorption peak at 492 nm and lutein showing an absorption peak at 460 nm, thus closer to the blue-violet light irradiation at 415 nm used in these experiments [27]. Such a protective effect of lutein and astaxanthin is also evident through their influence on the Nrf-2 pathway, as shown in Figures 5 and 6.

## 5. Conclusions

The results here illustrated show that both astaxanthin and lutein may protect the corneal epithelium from blue-violet light photo-oxidative and apoptotic damages, though with different efficacies. The association of the two molecules could be expected to be more protective versus white light damage because the two together would give a wider protection over the full length of the white light wavelengths, as we are presently investigating. The doses here used in the static in vitro model system are hardly correlated with the dynamic situation occurring in vivo and thus constitute only a proof of concept of the protective role that these two xanthophylls may have on the cells of the ocular surface.

**Author Contributions:** Conceptualization, M.C., C.D.A. and D.R.; Data curation, M.C., C.D.A. and G.S.; Formal analysis, M.C., C.D.A. and G.S.; Investigation, M.C., C.D.A. and G.S.; Methodology, M.C., C.D.A. and G.S.; Project administration, D.R.; Resources, D.R.; Software, M.C. and G.S.; Supervision, D.R. and G.L.; Validation, C.D.A. and G.L.; Writing—original draft, M.C. and D.R.; Writing—review and editing, M.C. and D.R. All authors have read and agreed to the published version of the manuscript.

**Funding:** This research received no external funding.

**Institutional Review Board Statement:** The protocol approval number by the Ethical Committee of the University Policlinic of Catania, Italy for the donation to research of a human cornea is prot. n. 58854 of 14 December 2021.

**Informed Consent Statement:** The HCE-F cell line was derived in our laboratory from an explanted cornea, obtained by the local ophthalmology clinic upon the informed consent of the patient and endorsement of the local ethic committee.

**Acknowledgments:** The technical assistance of the Catania university service center BRIT (Bio-nanotech Research and Innovation Tower) is highly acknowledged.

**Conflicts of Interest:** The authors declare no conflict of interest.

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
