# Peer review of "Comparative Efficiency of Lutein and Astaxanthin in the Protection of Human Corneal Epithelial Cells In Vitro from Blue-Violet Light Photo-Oxidative Damage"

_applsci, doi:10.3390/app12031268_

Round 1

Reviewer 1 Report

The manuscript named “Comparative efficiency of lutein and astaxanthin in the protection of human corneal epithelial cells from blue-violet light photo-oxidative damage.” is an interesting piece of work.

This work addresses the effect of exposure to blue-violet light and the protective role of lutein and astaxanthin on corneal cells. Please see my comments below.

The abstract is precise and meaningful.

The introduction provides a good background understanding. However, it can be improved by focusing on the objective of the work. It could be good to explain more about the topical formulation (reference 15) and the effects on the ocular surface.

The rest of- the content is well presented. The references are appropriate to the topic.

The presentation of data/information is clear and readable.

Figures are well arranged.

Discussion: Observations are well described with the support of scientific evidence. In my opinion, in the discussion should be indicated why the have selected the concentrations employed for the assays. It can be interesting a comparative of the amounts contained in the marketed formulation (reference 15) or other formulations in clinical trials.

In the discussion section, the lines 289-294 are not related with the aim of the work or at least, it is not clear the relation with the aim of the manuscript.  

Conclusion: Supported by results.

As minor comments, nowadays the l in the abbreviation for microliters must be capitalized. Please correct it in the manuscript. In addition, the measurement drops/mL (line 124) is not clear, and a clarification can be included.

Author Response

Comments and Suggestions for Authors

The manuscript named “Comparative efficiency of lutein and astaxanthin in the protection of human corneal epithelial cells from blue-violet light photo-oxidative damage.” is an interesting piece of work. Thank you!

This work addresses the effect of exposure to blue-violet light and the protective role of lutein and astaxanthin on corneal cells. Please see my comments below.

The abstract is precise and meaningful.

The introduction provides a good background understanding. However, it can be improved by focusing on the objective of the work. It could be good to explain more about the topical formulation (reference 15) and the effects on the ocular surface. The Introduction has been rearranged, also on suggestion of others referees, and a part from the discussion has been moved there (see lines 35-54 of the clean revised version). We have explained more in depth the results of the old reference 15 (now 26) (see lines 75-78 of the clean revised version).

The rest of- the content is well presented. The references are appropriate to the topic.

The presentation of data/information is clear and readable.

Figures are well arranged.

Discussion: Observations are well described with the support of scientific evidence. In my opinion, in the discussion should be indicated why the have selected the concentrations employed for the assays. It can be interesting a comparative of the amounts contained in the marketed formulation (reference 15) or other formulations in clinical trials. This is a study in vitro, so there cannot be a tight correlation with doses given in vivo. An explanation of the choice of doses in vitro is given in lines 198-200 of the clean revised version.

In the discussion section, the lines 289-294 are not related with the aim of the work or at least, it is not clear the relation with the aim of the manuscript.  This part has been moved to the Introduction (lines 35-54 of the clean revised version), following the indication of another referee.

Conclusion: Supported by results.

As minor comments, nowadays the l in the abbreviation for microliters must be capitalized. Please correct it in the manuscript. In addition, the measurement drops/mL (line 124) is not clear, and a clarification can be included. The notation has been corrected in the manuscript, though we believe that this is more an editorial choice.

Reviewer 2 Report

Carotenoids are recognized for their beneficial effect on health. The article “Comparative efficiency of lutein and astaxanthin in the protection of human corneal epithelial cells from blue-violet light photo-oxidative damage” presents an interesting and up to date study of the protective and antioxidant properties of lutein and astaxanthin on human primary corneal epithelial cells (HCE-F). Increasing the use of modern devices leads to increased exposure to blue-violet light and the eyes are most often exposed.

The article is well written, the methods are clearly described and also the results, however there should be made some corrections as:

12 – in vitro - should be written in italics in all the document (r54 also, r311, r 327, r340, r353, r369)

36-38 – can you add more references?

77 - In the materials and methods section details about the software and statistical methods used should be included

The discussions section is a little too general. Some of the information presented in this section I my opinion would rather fit in the introductory section, for example 287-297.

Discussions should be conducted a little more in the direction of comparing the results obtained in this article with other results obtained by other researchers in similar studies. Information about other studies are provided, but it does not seem to be sufficiently connected with the results obtained.

If this section is reformulated, and the results better related to the data of other researchers, I think it would add value to the article.

Author Response

Comments and Suggestions for Authors

Carotenoids are recognized for their beneficial effect on health. The article “Comparative efficiency of lutein and astaxanthin in the protection of human corneal epithelial cells from blue-violet light photo-oxidative damage” presents an interesting and up to date study of the protective and antioxidant properties of lutein and astaxanthin on human primary corneal epithelial cells (HCE-F). Increasing the use of modern devices leads to increased exposure to blue-violet light and the eyes are most often exposed.

The article is well written, the methods are clearly described and also the results, however there should be made some corrections as:

12 – in vitro - should be written in italics in all the document (r54 also, r311, r 327, r340, r353, r369) This has been corrected

36-38 – can you add more references? That has been done: now it is line 50 of the clean revised version and references 7-8-9-10 have been added in support.

77 - In the materials and methods section details about the software and statistical methods used should be included Done: see lines 173-176 of the clean revised version.

The discussions section is a little too general. Some of the information presented in this section I my opinion would rather fit in the introductory section, for example 287-297. We have rearranged the discussion, trying to make it more specific, and moved the indicated lines to the Introduction (lines 35-44 of the clean revised version).

Discussions should be conducted a little more in the direction of comparing the results obtained in this article with other results obtained by other researchers in similar studies. Information about other studies are provided, but it does not seem to be sufficiently connected with the results obtained. Most of the studies with carotenoids have considered only the retina, where they are accumulated and have a critical role for the protection of the structures. Only very few studies have considered the ocular surface, and this is why there is not much literature to cite, and this is also the novelty and the interest of our findings. However, we have added two more references in this part (lines 329-340 of the clean revised version, references 50 and 51).

If this section is reformulated, and the results better related to the data of other researchers, I think it would add value to the article.

Reviewer 3 Report

Title: This in an in vitro study so the title should not be misleading with the word ‘human’. Please revise.

Abstract: Please check grammar. ‘The aim of this study was to compare….’

Introduction:

Is it necessary to have figure in the introduction? Please retain the figure if only the compounds are newly identified. Also, for the spectrum figure.

Line 44: Lutein and zeaxanthin are already present in the human retina. So why is it important to conduct the present study? Please justify.

Line 71: What does it mean by ‘in association’? Does it mean in combination? Please clarify.

Materials and methods:

Line 79 & 80: They are not merely antioxidant molecules. They are carotenoids and might have other functions as well. Please revise.

Line 83: In relation to Line 44, how much lutein and zeaxanthin were retained during isolation? Please add info.

Line 87: Make it as CO2  

Line 89: Please add reference to this method.

Line 100: Please also add ref for this method. If this was a new method, please add the reproducibility data. This is very important for others to replicate the study.

Line 112: Please also add the reference for this method.

Results:

Line 171: Please check Figure 2. Something wrong with the layout.

Line 188: the figure is too small, and this should be improved. Need to specify what is SFM in the caption for Figure 3. Please indicate what is the number of replicates for each group in the caption.

Line 199: ‘Increase should be changed to increased.

 Discussion:

Need to add more info from animal of human studies.

 Conclusion:

 Need to add the word ‘in vitro’ in the conclusion since this is a cell culture study.

Author Response

Comments and Suggestions for Authors

Title: This in an in vitro study so the title should not be misleading with the word ‘human’. Please revise. We have added in the title the notation in vitro to be more explicit on this subject.

Abstract: Please check grammar. ‘The aim of this study was to compare….’ Corrected

Introduction:

Is it necessary to have figure in the introduction? Please retain the figure if only the compounds are newly identified. Also, for the spectrum figure. Figure 1 has been removed.

Line 44: Lutein and zeaxanthin are already present in the human retina. So why is it important to conduct the present study? Please justify. The present study addresses the potential protective role of carotenoids in cells of the ocular surface; nonetheless the addition of more carotenoids as food supplements appears to be important also for protection of retinal structures (see lines 68-71 and references 21-23 in the clean revised version).

Line 71: What does it mean by ‘in association’? Does it mean in combination? Please clarify. We have now used the suggested wording 'in combination'.

Materials and methods:

Line 79 & 80: They are not merely antioxidant molecules. They are carotenoids and might have other functions as well. Please revise. Done (see line 91 of the clean revised version).

Line 83: In relation to Line 44, how much lutein and zeaxanthin were retained during isolation? Please add info. The HCEF cell line used in these studies is a human corneal epithelial cell line, which has nothing to do with the retina. We did not and do not expect to find detectable amounts of carotenoids in the isolated epithelial cells from the human donor.

Line 87: Make it as CO2  Corrected

Line 89: Please add reference to this method. Done (ref. 28 in line 109 of the clean revised version).

Line 100: Please also add ref for this method. If this was a new method, please add the reproducibility data. This is very important for others to replicate the study. Done (ref. 30-31 in line 119 of the clean revised version)

Line 112: Please also add the reference for this method. Done (ref. 32 in line 126 of the clean revised version).

Results:

Line 171: Please check Figure 2. Something wrong with the layout. We checked and do not see anything wrong. Something might have happened with the downloading of the file.

Line 188: the figure is too small, and this should be improved. Need to specify what is SFM in the caption for Figure 3. Please indicate what is the number of replicates for each group in the caption. We have added in each figure legend the explanation of SFM (Serum Free Medium) and the number of replicates. 

Line 199: ‘Increase should be changed to increased. Now the incriminated word is in line 119: "As expected, the absence of any protection resulted in a 79% decrease in cell viability (Figure 3A), and a 394% increase in ROS production" . We do not think it should be changed as indicated.

 Discussion:

Need to add more info from animal of human studies. There is not much work done in animals or humans on carotenoids protection of the ocular surface. We have added two more references (50 and 51) to enrich this part.

 Conclusion:

 Need to add the word ‘in vitro’ in the conclusion since this is a cell culture study. Done: see lines 425-428 of the clean revised version of the manuscript.

Round 2

Reviewer 2 Report

The author took into account the recommendations and significantly improved the quality of the article, in my opinion the article can be accepted for publication in the present form.

Author Response

The last required additions have now been made:

Line 75 reports the level of significance of the statistical test 

Lines 443-445 report the Institutional Review Board Statement 
